# Antibacterial Activity of Kaolin–Silver Nanomaterials: Alternative Approach to the Use of Antibiotics in Animal Production

**DOI:** 10.3390/antibiotics10111276

**Published:** 2021-10-20

**Authors:** Lara Pérez-Etayo, David González, José Leiva, María Díez-Leturia, Alba Ezquerra, Luis Lostao, Ana Isabel Vitas

**Affiliations:** 1Department of Microbiology and Parasitology, University of Navarra, 31008 Pamplona, Spain; dgonzalez@unav.es (D.G.); mdiezlet@unav.es (M.D.-L.); avitas@unav.es (A.I.V.); 2IdiSNA, Navarra Institute for Health Research, 31008 Pamplona, Spain; 3Microbiology Service, Clínica Universidad de Navarra, University of Navarra, 31008 Pamplona, Spain; jleiva@unav.es; 4ENOSAN Laboratories, 50018 Zaragoza, Spain; investigacion@laboratoriosenosan.com (A.E.); presidente@laboratoriosenosan.com (L.L.)

**Keywords:** antimicrobial resistance, animal production, additives, feed, nanomaterials, silver

## Abstract

According to the search for alternatives to replace antibiotics in animal production suggested in the antimicrobial resistance action plans around the world, the objective of this work was to evaluate the bactericidal effect of kaolin–silver nanomaterial for its possible inclusion as an additive in animal feed. The antibacterial activity of the C3 (kaolin–silver nanomaterial) product was tested against a wide spectrum of Gram-negative and Gram-positive bacteria (including multidrug resistant strains) by performing antibiograms, minimal inhibitory concentration (MIC) and minimal bactericidal concentration (MBC), as well as growth inhibition curves against seven strains causing infections in animals. The C3 product generated inhibition halos in all the tested strains, and a higher activity against Gram-negative bacteria was found, with MBC values ranged from 7.8 µg/mL (*P. aeruginosa*) to 15.6 µg/mL (*E. coli* and *Salmonella*). In contrast, it was necessary to increase the concentration to 31.3 µg/mL or 250 µg/mL to eliminate 99.9% of the initial population of *S. aureus* ATCC 6538 and *E. faecium* ATCC 19434, respectively. Conversely, the inhibition growth curves showed a faster bactericidal activity against Gram-negative bacteria (between 2 and 4 h), while it took at least 24 h to observe a reduction in cell viability of *S. aureus* ATCC 6538. In short, this study shows that the kaolin–silver nanomaterials developed in the framework of the INTERREG POCTEFA EFA183/16/OUTBIOTICS project exhibit antibacterial activity against a wide spectrum of bacteria. However, additional studies on animal safety and environmental impact are necessary to evaluate the effectiveness of the proposed alternative in the context of *One Health*.

## 1. Introduction

The use of antibiotics in animal feed for prophylactic and therapeutic purposes implies the use of subtherapeutic concentrations during prolonged exposure times, which favor the appearance and spread of antibiotic resistant bacterial strains [1]. In this way, animals act as a reservoir for genes and resistant bacteria that can be transmitted to humans through food [2,3]. Thus, Calbo et al. [4] determined the implication of a multi-resistant strain (MDR) of *K. pneumoniae* producing extended spectrum beta-lactamase (ESBL) in a nosocomial foodborne outbreak in Spain, demonstrating that food can be a vector for the transmission of ESBL. Other studies showed that the percentage of quinolone-resistant *E. coli* strains isolated from chickens and pigs in some European Union countries (Spain, Poland, United Kingdom, Germany, and France) is high, suggesting that these antibiotics have been widely used prophylactically in these countries [3]. Similarly, an outbreak of salmonellosis in the United States (USA) was related to chickens contaminated with a strain of *Salmonella* MDR [5]. In short, exposure to antibiotics in animal and human environments lead to an increase in the incidence of resistant bacteria [6,7]. For this reason, the European Union (EU) banned the use of antibiotics as growth promoters in feed in 2006 [8]. However, despite these prohibitions and according to the European Medicines Agency (EMA), between 2010 and 2015, Spain was the second country where more antibiotics were used in livestock farming (402 mg/kg of meat produced) [9]. Fortunately, after the implementation of the National Antibiotic Resistance Plan (PRAN) in June 2014, there was a 32.4% reduction in the consumption of antibiotics in animal health in the 2014–2017 period, showing the effectiveness of the REDUCE programs established in the different livestock sectors. For instance, this initiative has managed to reduce the consumption of colistin in pigs by 97.2%, and in poultry, the total consumption of antibiotics has been reduced by 71% in recent years [10].

Given that, the use of zootechnical additives in animal feed is a common practice and due to the prohibition of the use of antibiotics as growth promoters, the search for alternatives to the use of antibiotics continues. Current options most used in poultry and swine production include the use of organic acids, probiotics, prebiotics, essential oils, plant extracts and metals such as copper and zinc [11]. However, although the organic forms of these last elements are still greater as food additives in animal feed [12], the inorganic forms have disadvantages such as considerable tissue retention and being potentially contaminating [13]. Following this line of action, silver was used as an additive in chicken feed in the 1950s, but due to its high manufacturing cost, it was no longer used [14]. Currently, and due to the technological development of industrial nanoparticle manufacturing processes, silver nanoparticles (AgNPs) are of special interest [15]. The emergence of nanotechnology has created opportunities to explore the antibacterial effects of metal nanoparticles and several authors have shown that silver nanoparticles have higher antibacterial activity compared to other metallic nanoparticles, even against multidrug resistant bacteria [16,17]. In addition, Fondevila et al. [18] observed that silver tissue retentions in piglets were 100 times lower than those referred to by the inclusion of zinc. At the same time, the contaminating potential of silver excretion through slurry is considered low compared to zinc and copper [18]. Therefore, this technology means that AgNPs can be proposed as a possible alternative in the use of antimicrobial additives for animal feed.

For centuries, silver has been known for its antibacterial effects and has been used to prevent and control infections caused by microorganisms due to its low cytotoxicity [19]. The mode of action of this metal has not yet been fully elucidated and according to various studies, it exerts its antimicrobial activity through different mechanisms: interacts with the bacterial cell surface, altering the functions of cell membranes and causing structural changes that cause bacteria are more permeable [15]; it reacts with bacterial DNA, inhibiting cell replication and causing bacterial death [20]. At the same time, the fact that bacterial resistance to this element is extremely rare [21,22], gives it a special advantage for the development of antibacterial alternatives that contain this material [23]. Conversely, clays are also used in animal feed for nutritional purposes (increased digestion of nutrients, reduced traffic speed), health (prevention of diarrhea) and environmental (reduction of ammonia emissions and bad odors) [24]. They have the ability to adsorb toxins produced by bacteria and reducing bacterial adhesion to the surface of epithelial cells [25]; thus, a compound based on clays and silver nanoparticles may have a synergistic effect.

One of the objectives of the INTERREG POCTEFA EFA183/16/OUTBIOTICS project [26] is to evaluate the efficacy and safety of compounds based on clay and silver nanoparticles, as a possible alternative for their use as additives in animal feed, guaranteeing safety for animals and the environment. This approach is in accordance with Regulation (EC) No. 1831/2003 [27] on additives in animal feed, which includes “the need to develop alternative products that replace the use of antibiotics”. In this way, the specific objective of this work has been to evaluate the bactericidal activity of the C3 material (silver nanoparticles deposited on a kaolinite base material) developed by Laboratories Enosan S.L. against a selection of Gram-positive and Gram-negative bacteria, including bacteria resistant to different groups of antibiotics.

## 2. Results

This study was aimed to determine the antibacterial effect of kaolin–silver nanomaterials against a selection of strains (characteristics described in material and methods section). The results of the inhibition halos obtained with the different batches of C3 product, C2 product and control antibiotics, are shown in Table 1.

The presence of clear zone around the AgNPs wells suggests that the C3 nanomaterial (all batches) has antibacterial activity against all the tested strains. Thus, it is able to inhibit the growth of the ESKAPE and resistant strains, generating inhibition halos with diameters ranging from 11–21 mm (Table 1). On the contrary, the product without silver (C2) exhibited no activity (the value 6 mm corresponds to the well diameter). The inhibition halos produced by control antibiotics were in accordance with those published by CLSI for the different bacterial families [28]. As an example, Figure 1 illustrates some antibiograms obtained against sensitive Gram-negative strain (Figure 1A) and against antibiotic resistant Gram-positive strain (Figure 1B).

Antibiograms test was described as the preliminary study in screening the antibacterial activity of an antibacterial agent. For that reason, a further evaluation in determining the antibacterial activity of AgNPs by means of minimum inhibitory concentration (MIC) and minimum bactericidal concentration (MBC) values was required. In this sense, Table 2 summarizes the results obtained with the 15 strains selected for this study (sensitive ones and representative of each type of resistance). The MIC and MBC values correspond to the theoretical µg/mL of silver in the well according to the percentage of silver present in the nanomaterial. MIC and MBC tests were performed in triplicate for each of the strains.

As we can see, the C3 product has higher activity against Gram-negative bacteria and no differences have been observed between antibiotic resistant and sensitive strains, except for *S. aureus* MRSA 2. Thus, MBC against Gram-negative bacteria ranged from 7.8 µg/mL (*P. aeruginosa*) to 15.6 µg/mL (*E. coli* and *Salmonella*). However, it was necessary to increase the concentration to 31.3 µg/mL or 250 µg/mL to obtain the MBC in the case of Gram-positive bacteria such as *S. aureus* ATCC 6538 and *E. faecium* ATCC 19434, respectively.

The inhibition growth curves obtained against the seven selected strains are shown in the Figure 2, Figure 3 and Figure 4. In the tests with *E. coli* ATCC 8739 and *S. aureus* ATCC 6538, the product C2-L2 (50 and 100 mg/mL, respectively) was also included as a control (Figure 2A and Figure 4A). The absence of antibacterial activity of the silver-free kaolin support material was confirmed, since the curves are coincident to those obtained with the positive control growth (no significant differences in all the points).

It is observed that the bactericidal activity is faster and with lower silver concentration in the case of *E. coli* ATCC 8739, with a reduction of 100% of the initial microorganisms in 8 h with 7.8 µg/mL of the product (*p* < 0.0001 compared with positive control, Figure 2A). In addition, the 15.6 and 31.3 µg/mL concentrations produced a reduction of more than 3 log_10_ CFU/mL in the bacterial count after 4 h of incubation (statistically significant differences compared with positive control, *p* < 0.0001) and this lethal effect persisted during the 24 h duration of the experiment. This effect was also observed with the *E. coli* ESBL strain 3 isolated from a farm (*p* < 0.0001, Figure 2C), unlike what happened with the *E. coli* ESBL strain 2 (Figure 2B), in which 24 h of incubation were necessary to obtain a similar reduction with both concentrations (*p* < 0.0001 compared with positive control). *Salmonella typhimurium* and *Salmonella enteritidis* shown a similar behavior, as can be seen in Figure 3A,B. At 2 h of incubation, the bactericidal effect is observed with the concentrations 15.6 and 31.3 µg/mL (statistically significant differences compared with positive control, *p* < 0.0001).

Finally, the inhibition growth curves with the Gram-positive strains (Figure 4), confirmed that both are more resistant to the antibacterial action of the C3 nanomaterial, with a higher resistance being observed with the clinical *S. aureus* MRSA 2 strain, positive for the *mecA* gene (Figure 4B). Specifically, with the *S. aureus* ATCC 6538 strain, the bactericidal effect with logarithmic reduction was observed after 24 h of incubation with 31.3 and 62.5 µg/mL (statistically significant differences compared with positive control, *p* < 0.0001), while with *S. aureus* MRSA 2 this effect was not achieved.

## 3. Discussion

In order to achieve the objective proposed in this work, different methodologies were performed. The antibiograms carried out showed the bactericidal activity in all the strains tested (Gram-negative and Gram-positive strains, susceptible or resistant to antibiotics), which suggests that these silver nanoparticles are broad spectrum antibacterial agents. These results are in accordance with previous studies, where it was found that silver nanoparticles exert an effect on different groups of bacteria [29,30]. In agreement with our results, other works show that silver nanoparticles are effective against bacteria resistant to antibiotics such as *E. coli* and *P. aeruginosa* [31,32], *S. aureus* resistant to methicillin (MRSA) [33] and ESBL-producing enterobacteria [34]. Similarly, Kim et al. [35] studied the antimicrobial activity of some AgNPs against *E. coli* and *S. aureus* and showed that *E. coli* was inhibited at lower concentrations, while the inhibitory effects on the growth of *S. aureus* were less marked. Similarly, our results show a lower MBC of product C3 against Gram-negative bacteria (15.6 µg/mL for *E. coli* and *Salmonella* and 7.8 µg/mL for *P. aeruginosa*), while a higher concentration of product is required to destroy the 99.9% of the initial population of Gram-positive bacteria (31.3 and 250 µg/mL for *S. aureus* ATCC 6538 and *E. faecium* ATCC 19434, respectively). Conversely, the results obtained by Loo et al. [36] with another type of silver nanoparticles, show a lower MBC for *E. coli* and *Salmonella*, while these concentrations are higher with the nanoparticles developed by Lara et al. [29]. Although the comparison of results on MIC and MBC is difficult, since there is no standard method or cut-off points for determining the antibacterial activity of the different AgNPs developed, the fact is that the observed trends are similar and point toward greater activity against Gram-negative [37]. Despite the differences found with the MIC/MBC, these results also correlate with the inhibition curves obtained in a liquid medium, since this methodology is considered more sensitive, because it is carried out in greater volume and the microorganisms are in continuous contact with the nanoparticles due to the constant agitation. Thus, more time was needed to observe the bactericidal effect of the C3 product on the cell viability of *S. aureus* ATCC 6538 and *S. aureus* MRSA 2. This could be due to the structural difference in the composition of the cell walls of both types of bacteria, specifically, to the way in which AgNPs interact with the cell wall [15]. In the same line, a study with *E. coli* confirmed that the accumulation of silver nanoparticles in the cell membrane creates spaces in the bilayer, leading to greater permeability and finally bacterial cell death [20]. On the contrary, the wall of Gram-positive bacteria could generate a greater resistance to the entrance of nanoparticles [38]. Thus, a thicker peptidoglycan layer would act as a physical barrier against AgNPs, regardless of the mechanism of action involved, as shown in Figure 5.

Moreover, the C3 material was able to destroy a high concentration of bacteria (approximately 10^5^–10^6^ CFU/mL) in relatively low concentrations of silver (between 7.8 and 15.6 µg/mL for Gram-negative and 31.3 µg/mL for Gram-positives), after 24 h of shaking exposure. Therefore, these results indicate that the kaolin–silver nanomaterial developed in this project shows an excellent antibacterial effect, in accordance with the reported by Loo et al. [36]. The novelty of this work is to establish the bases for the development of an antibacterial based on kaolin–silver nanomaterials, as an alternative to the use of antibiotics in livestock. Current technology allows the production of these silver nanoparticles at a very reasonable cost, using low concentrations of the metal. Although the antibacterial activity of silver is well known [20], it was necessary to demonstrate that the nanomaterial C3 developed in this project with 1% silver, is active against bacteria that cause the most common pathologies in animal production, especially against resistant bacteria classified by the WHO as of critical and high priority [39].

Some authors have suggested that feeding with silver nanoparticles improves digestive efficiency, immunity and performance in livestock and poultry [40], although studies on the effects in production animals are limited. Therefore, the introduction of these novel materials in animals and consumer products requires safety assessments, as well as a stronger understanding of any potential impacts on both human health and environment. In this way, Abad-Álvaro et al. [24], studied the effects on tissue silver retention in pigs, using kaolin–silver nanoparticles similar to product C3. The results showed that silver accumulated in the liver during the feeding period with the feed supplemented with the nanomaterial was low in comparison to the levels of silver excreted in feces. Conversely, silver was not found in significant values in the muscle tissues, which makes this technology a potential alternative as a growth promoter. However, the environmental impact of the feces must be evaluated, which could limit its application to the soil as a manure management practice.

## 4. Materials and Methods

### 4.1. Products

The products used in the different in vitro tests were supplied by Laboratories Enosan S.L. The silver-based nanomaterial preparation treatment allows the deposit of silver on the kaolin surface in the form of metallic silver nanoparticles. Four batches of product C3 (L1, L6, L7 and L8), containing approximately 1% silver, were analyzed. The inert material C2 (L2) was used as a control (kaolin without silver, with technological treatment similar to C3).

To test the bactericidal activity of the test products, suspensions of 50 mg/mL or 100 mg/mL were prepared in type II deionized water (Wasserlab, Millipore, Barbatáin, Navarra). The suspensions were prepared the same day the test was performed and were kept at room temperature.

### 4.2. Agar Well-Diffusion Antibacterial Assay

This part of the study was carried out with the 23 Gram-negative and Gram-positive bacteria shown in Table 3. Reference strains coming from the Spanish Collection of Cultures Type (CECT) with antibiotic-sensitive phenotypes (with the exception of *Klebsiella pneumoniae* CECT 7787) are included, among which are species classified as especially resistant to antibiotics (ESKAPE; *Enterococcus faecium*, *Staphylococcus aureus*, *Klebsiella pneumoniae*, *Pseudomonas aeruginosa* and *Enterobacter*). In addition, two species of *Salmonella* are included, given their relationship with infections transmitted in the animal environment. Finally, antibiotic resistant bacteria isolated from different aquatic and farm environments from POCTEFA territory [41] and clinical origin completed the strain selection. All resistant strains were previously identified and characterized phenotypically and genotypically, to know the pattern of susceptibility to antimicrobials, the types of β-lactamase and the genes involved in resistance to methicillin (*mecA*), colistin (*mcr*) and vancomycin (*vanB*) [41].

From the set of frozen strains (−80 °C) the working strains were prepared by spreading in general media (TSA; Scharlau, Sentmenat, Barcelona) and incubating at 37 °C for 24 h. The strains were kept refrigerated (5 °C ± 3 °C), making periodic passages to fresh media for the different tests (incubation at 37 °C for 24 h). From these fresh cultures, inocula were prepared in peptone broth (APT; Scharlau) adjusting the concentration in a densitometer (DEN-1-BIOSAN) equivalent to 0.5 McFarland (approximately 1.5 × 10^8^ CFU/mL). After spreading the inoculum with a swab on the surface of a cation-adjusted Mueller Hinton II agar plate (MH, Becton Dickinson, Le Pont de Claix, France), holes of 6–7 mm diameter were done and 50 µL of the suspension of the product to be tested (50 mg/mL) was added. Antibiotic discs were used as positive inhibition controls: Ampicillin 10 µg (AMP), Amoxicillin/Clavulanate 30 µg (AMC) and Gentamicin 10 µg (GM), supplied by Becton Dickinson. The plates were incubated at 37 °C for 24 h and subsequently the diameter of the generated inhibition halos was measured.

### 4.3. Minimal Inhibitory Concentration (MIC) and Minimal Bactericidal Concentration (MBC) Determination

MIC and MBC of the product C3-L1 was determined against the 15 strains shown in Table 2, including sensitive and resistant ones. The tests were carried out by the microdilution technique in 96-well plates. Given the turbidity provided by the product itself in suspension, resazurin was added to calculate the MIC, due to the color change that occurs when there is growth of microorganisms (from the natural color purple to pink in the reduced form). Resazurin (Acros organics, Waltham, MA, USA) was prepared at 0.02%, dissolving 0.002 g in 10 mL of deionized H_2_O type II, homogenizing with a stirrer and sterilizing through a 0.22 µm filter (Millipore Millex PES).

A fresh culture of the strains to be tested was prepared on TSA agar (24 h, 37 °C). After adjusting the inoculum to 0.5 on the McFarland scale, a 1/100 dilution of the inoculum was made in MH broth. To know the initial concentration of each microorganism, serial dilutions and plating on TSA agar were made. Suspensions of the C3-L1 product were prepared at 50 or 100 mg/mL, for Gram-negative or Gram-positive bacteria, respectively (silver concentration equivalent to 500 or 1000 µg/mL). After adding 100 µL of the C3 product suspension to the first well, 8 double dilutions were made (previously, 100 µL of sterile MH broth was added to each well). Then, 30 µL of the resazurin solution and 100 µL of the prepared inoculum were added to each well (initial concentration in each well around 1.5 × 10^5^ CFU/mL). Each of the concentrations was tested in triplicate. The last two columns corresponded to negative and positive controls.

The plates were incubated at 37 °C for 24 h. The MIC corresponded to the concentration of the last well in which there was no color change. MBC was determined by seeding 0.1 mL from the well determined as MIC and higher concentrations on TSA agar. After incubation at 37 °C for 24 h, the CFUs were counted. Taking into account the initial count of each microorganism, the MBC corresponded with the concentration that eliminates the 99.9% of initial population.

### 4.4. Growth Inhibition Curves in Liquid Medium

The antibacterial activity of the C3-L1 nanomaterial as a function of time was studied against 7 strains (*E. coli* ATCC 8739, *E. coli* ESBL 2, *E. coli* ESBL 3, *S. typhimurium* ATCC 14028, *S. enteritidis* ATCC 13076, *S aureus* ATCC 6538 and *S. aureus* SARM 3). Growth inhibition curves were performed in liquid medium (MH broth) using different concentrations (MBC, MIC and sub-inhibitory concentrations), using sterile tubes which contained 4 mL of sterile MH broth, 1 mL of the C3-L1 product solution and 50 µL of bacterial inoculum (0.5 McFarland adjusted concentration). In addition, 2 control tubes were prepared containing 1 mL of C2-L2 solution (negative inhibition control) and 1 mL of deionized water (positive growth control) instead of the C3-L1 product. Immediately after the inoculation of the different tubes, the microorganism count was carried out to know the initial starting concentration (CFU/mL). Then, the tubes were incubated at 37 °C with shaking and samples were taken at different times (1, 2, 4, 8 and 24 h) for subsequent enumerations. Serial dilutions were made, and 0.1 mL of each dilution was seeded on TSA agar, to know the number of viable microorganisms. Subsequently, the log_10_ CFU/mL was calculated and the different inhibition curves were represented graphically.

### 4.5. Statistical Analysis

The results of the growth inhibition curves were subjected to statistical processing with GraphPad Prism Software (GraphPad Software Inc., San Diego, CA, USA), applying the t-student test with a level of significance of *p* < 0.05.

## 5. Conclusions

In conclusion, the kaolin–silver nanomaterial (C3) developed in the framework of the INTERREG POCTEFA EFA183/16/OUTBIOTICS project showed excellent antibacterial activity against a wide spectrum of bacteria, including MDR strains. Therefore, from a microbiological point of view, this product meets the purpose of being a good candidate as a zootechnical additive in animal feed. If the biosafety premises that were carried out in the aforementioned project are met (toxicological studies, distribution in biological samples, environmental impact and economic viability), these materials could be a good candidate to replace or reduce, as much as possible, the use of antibiotics in animal production.

## 6. Patents

Feed additive for animals: national patent 200701496 and Nanosystems comprising silver and antibiotics and their use for the treatment of bacterial infections. Patent in international extension (China, Japan, USA, EPO, Brazil, Colombia and Mexico) PCT/EP2018/059006. 

## Figures and Tables

**Figure 1 antibiotics-10-01276-f001:**
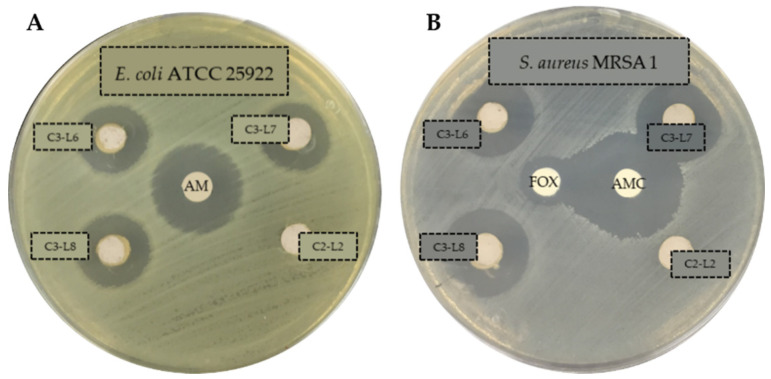
Antibiograms of C3 (with silver, batches L6, L7 and L8) and C2 (without silver, batch L2) products against *E. coli* ATCC 25922 (**A**) and *S. aureus* MRSA 1 (**B**). C3 and C2 products in solution 50 mg/mL. Ampicillin (AM, 10 µg), Amoxicillin/Clavulanate (AMC, 30 µg), cefoxitina (FOX, 30 µg).

**Figure 2 antibiotics-10-01276-f002:**
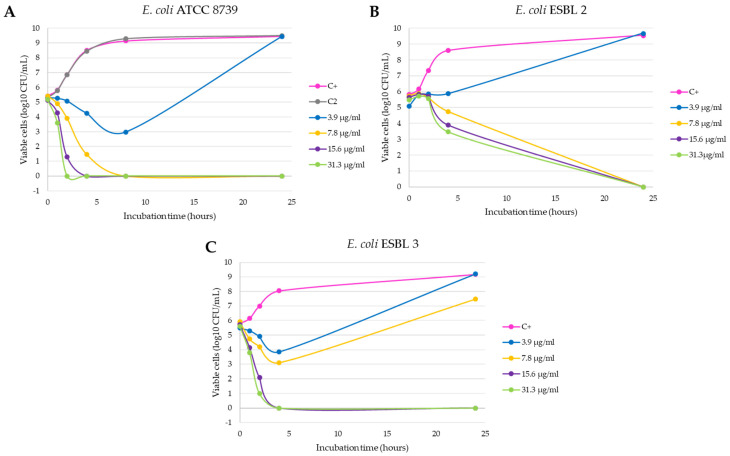
Inhibition curves of product C3-L1 in liquid medium against the Gram-negative strains *E. coli* ATCC 8739 (**A**), *E. coli* ESBL 2 (**B**) and *E. coli* ESBL 3 (**C**).

**Figure 3 antibiotics-10-01276-f003:**
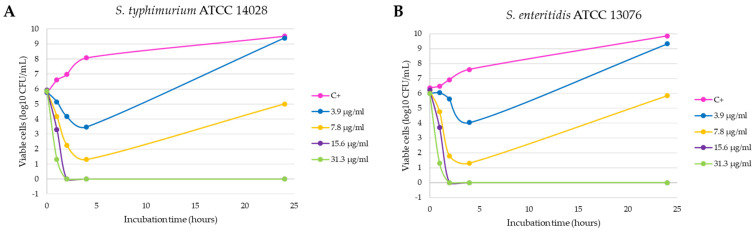
Inhibition curves of product C3-L1 in liquid medium against the Gram-negative strains *Salmonella typhimurium* ATCC 14028 (**A**) and *Salmonella enteritidis* ATCC 13076 (**B**).

**Figure 4 antibiotics-10-01276-f004:**
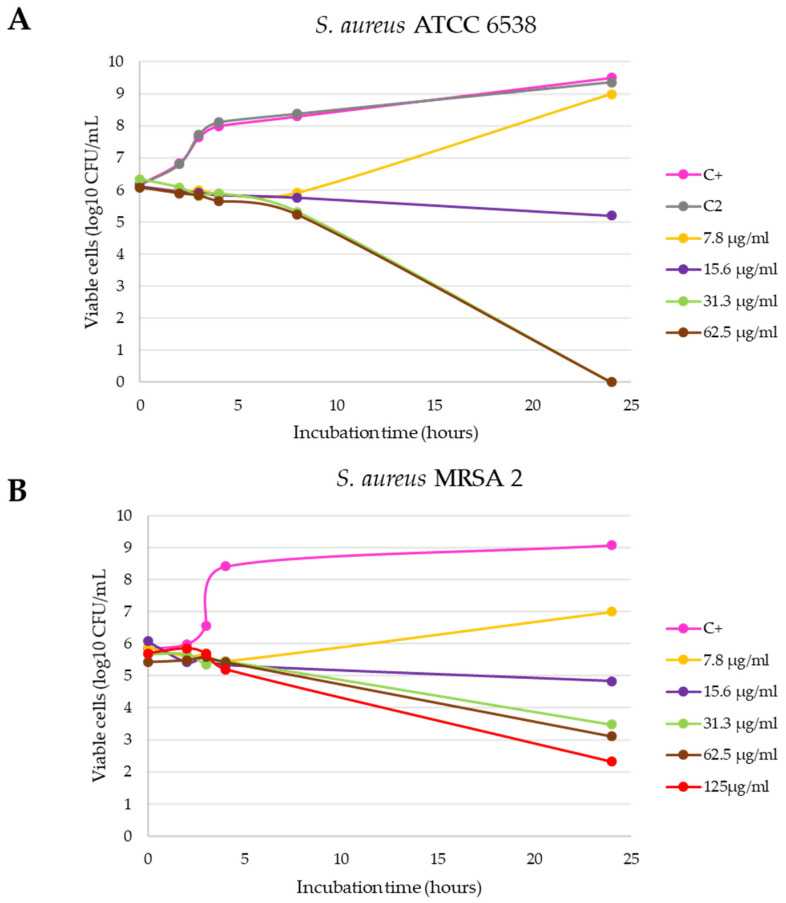
Inhibition curves of the C3-L1 product in liquid medium against *S. aureus* ATCC 6538 (**A**) and *S. aureus* MRSA 2 (**B**).

**Figure 5 antibiotics-10-01276-f005:**
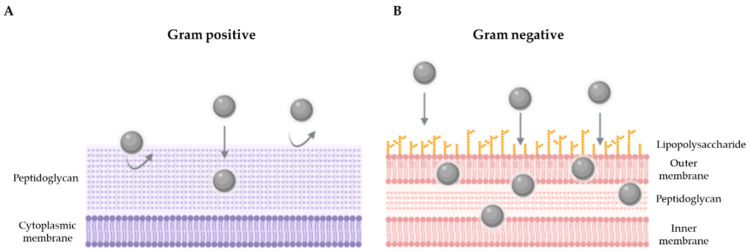
Hypothetical model showing the interaction of AgNPs against Gram-positive (**A**) and Gram-negative (**B**) bacteria.

**Table 1 antibiotics-10-01276-t001:** Inhibition halos of the products C2 (without silver), C3 (with silver) and control antibiotics (diameter in mm).

Strain	C2 ^1^	C3 ^1^				Control Antibiotics ^2^	
	L2	L6	L7	L8	X ± sd		
*E. faecium* ATCC 19434	6	19	20	21	20 ± 1	AMC	37
*S. aureus* ATCC 25923	6	20	20	20	20 ± 0	GM	24
*K. pneumoniae* ATCC 700603	6	15	15	15	15 ± 0	AMC	19
*A. baumannii* ATCC 19606	6	19	20	20	19.67 ± 0.58	GM	17
*P. aeruginosa* ATCC 27853	6	17	18	17	17.33 ± 0.58	GM	17
*E. cloacae* ATCC 13047	6	12	11	12	11.67 ± 0.58	GM	20
*E. coli* ATCC 25922	6	17	16	16	16.33 ± 0.58	AM	20
*Salmonella enteritidis* ATCC 13076	6	15	15	15	15 ± 0	AM	27
*Salmonella typhimurium* ATCC 14028	6	15	16	15	15.33 ± 0.58	AM	26
*E. coli* ESBL 1	6	15	15	14	14.67 ± 0.58	AMC	22
*E. coli* ESBL 2	6	16	16	16	16 ± 0	AMC	19
*E. coli* ESBL 3	6	18	18	18	18 ± 0	AMC	20
*S. fonticola* ESBL	6	15	16	16	15.67 ± 0.58	AMC	19
*K. pneumoniae* ESBL	6	14	14	15	14.33 ± 0.58	AMC	19
*E. coli* CARBA	6	15	15	15	15 ± 0	GM	21
*C. freundii* CARBA	6	17	17	17	17 ± 0	GM	14
*P. aeruginosa* CARBA 1	6	21	21	21	21 ± 0	GM	21
*P. aeruginosa* CARBA 2	6	20	20	21	20.33 ± 0.58	GM	19
*E. coli* COL	6	16	16	16	16 ± 0	AMC	22
*K. oxytoca* COL	6	17	18	17	17.33 ± 0.58	AMC	30
*S. aureus* MRSA 1	6	20	20	19	19.67 ± 0.58	AMC	25
*S. aureus* MRSA 2	6	18	19	18	18.33 ± 0.58	GM	25
*E. faecium* VANCO 2	6	20	20	21	20.33 ± 0.58	AMC	40

^1^ C2: batch L2; C3: batches L6, L7, and L8. Value 6 means absence of halo (6 mm is the diameter of the well). ^2^ Ampicillin (AM, 10 µg), Gentamicin (GM, 10 µg), Amoxicillin/Clavulanate (AMC, 30 µg).

**Table 2 antibiotics-10-01276-t002:** MIC and MBC values (µg/mL silver) for the C3-L1 product against various strains. Identical results were obtained in the three replicates.

Strain	MIC (µg/mL)	MBC (µg/mL)
*E. coli* ATCC 8739	7.8	15.6
*E. coli* ESBL 1	7.8	15.6
*E. coli* ESBL 2	7.8	15.6
*E. coli* ESBL 3	7.8	15.6
*E. coli* CARBA	7.8	15.6
*E. coli* COL	7.8	15.6
*C. freundii* CARBA	7.8	15.6
*Salmonella. enteritidis* ATCC 13076	7.8	15.6
*Salmonella typhimurium* ATCC 14028	7.8	15.6
*P. aeruginosa* ATTC 9027	3.9	7.8
*P. aeruginosa* CARBA 2	3.9	7.8
*S. aureus* ATCC 6538	15.6	31.3
*S. aureus* MRSA 2	15.6	125
*E. faecium* ATCC 19434	7.8	250
*E. faecium* VANCO 1	7.8	250

**Table 3 antibiotics-10-01276-t003:** Strains used in agar well-diffusion antibacterial assays and their characteristics.

Strain	Origin ^1^	Resistance	Resistance Gene
*Enterococcus faecium* ATCC 19434	CECT (410)	-	-
*Staphylococcus aureus* ATCC 25923	CECT (435)	-	-
*Klebsiella pneumoniae* ATCC 700603	CECT (7787)	Penicillins	*bla* _SHV-18_
*Acinetobacter baumannii* ATCC 19606	CECT (9111)	-	-
*Pseudomonas aeruginosa* ATCC 27853	CECT (108)	-	-
*Enterobacter cloacae* ATCC 13047	CECT (194)	-	-
*Escherichia coli* ATCC 25922	CECT (434)	-	-
*Salmonella enteritidis* ATCC 13076	CECT (4300)	-	-
*Salmonella typhimurium* ATCC 14028	CECT (4594)	-	-
*Escherichia coli* ESBL 1	River	Penicillins/cephalosporins	*bla*_CTX-M14_, *bla*_TEML-278_
*Escherichia coli* ESBL 2	River	Penicillins/cephalosporins	*bla*_SHV-12_, *bla*_TEML-278_
*Escherichia coli* ESBL 3	Pig farm	Penicillins/cephalosporins	ND
*Serratia fonticola* ESBL	River	Penicillins/cephalosporins	*bla* _CTX-M1_
*Klebsiella pneumoniae* ESBL	WWTP	Penicillins/cephalosporins	*bla* _CTX-M14_
*Escherichia coli* CARBA	WWTP	Penicillins/cephalosporins/ carbapenems	*bla*_TEML-278_, KPC
*Citrobacter freundii* CARBA	WWTP	Penicillins/cephalosporins/ carbapenems	*bla*_TEML-278_, KPC
*Pseudomonas aeruginosa* CARBA 1	WWTP	Penicillins/cephalosporins/ carbapenems	*bla* _TEML-278_
*Pseudomonas aeruginosa* CARBA 2	WWTP	Penicillins/cephalosporins/ carbapenems	ND
*Escherichia coli* COL	Rabbit farm collector	Colistin	*mcr-1*
*Klebsiella oxytoca* COL	River	Colistin	ND
*Staphylococcus aureus* MRSA 1	River	Methicillin	ND
*Staphylococcus aureus* MRSA 2	Clinical sample	Methicillin	*mecA*
*Enterococcus faecium* VANCO 2	Rabbit farm collector	Vancomycin	*vanB*

^1^ WWTP: wastewater treatment plant; ND: not detected.

## Data Availability

Not applicable.

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
