# Peer review of "Antibacterial Activity of Kaolin–Silver Nanomaterials: Alternative Approach to the Use of Antibiotics in Animal Production"

_antibiotics, 2021, doi:10.3390/antibiotics10111276_

Round 1

Reviewer 1 Report

Dear authors.

This manuscript reports the results of tests carried out in vitro on the antibacterial properties of silver nanoparticles (AgNPs). The methodology, results and discussion are in accordance with the objectives proposed by the authors. In this sense, the manuscript is suitable to present it in the approach of the journal "Antibiotics". However, I must make three comments:

(1) The term "antimicrobial activity" is not used properly, it should be antibacterial activity, since only bacteria were tested (gram positive and gram negative). Antimicrobial activity involves testing on other microbes (fungi, riketsiae, protozoos and even viruses). 

I suggest you change the title of your study on microbial activity to ANTIBACTERIAL ACTIVITY OF KAOLIN-SILVER NANOPARTICLES ..., since only bacteria have been assayed on ...

(2) For the same reason, to date, there are already scientific reports that mention the antiviral and antifungal activity of AgNPs.

(3) Despite my comments, the journal "Antibiotics" could limit the terminology or allow the publication of this manuscript if an agreement with other authors has already been established in previous articles.   

Author Response

Dear authors.

This manuscript reports the results of tests carried out in vitro on the antibacterial properties of silver nanoparticles (AgNPs). The methodology, results and discussion are in accordance with the objectives proposed by the authors. In this sense, the manuscript is suitable to present it in the approach of the journal "Antibiotics". However, I must make three comments:

(1) The term "antimicrobial activity" is not used properly, it should be antibacterial activity, since only bacteria were tested (gram positive and gram negative). Antimicrobial activity involves testing on other microbes (fungi, riketsiae, protozoos and even viruses).

I suggest you change the title of your study on microbial activity to ANTIBACTERIAL ACTIVITY OF KAOLIN-SILVER NANOPARTICLES ..., since only bacteria have been assayed on ...

(2) For the same reason, to date, there are already scientific reports that mention the antiviral and antifungal activity of AgNPs.

(3) Despite my comments, the journal "Antibiotics" could limit the terminology or allow the publication of this manuscript if an agreement with other authors has already been established in previous articles

 Thank you very much for your comments. We are very grateful to the reviewer for the suggestions made and in order to clarify theses aspects, we have changed the title of the manuscript “Antibacterial activity of kaolin-silver nanomaterials: alternative approach to the use of antibiotics in animal production”. In addition, the text has been clarified according to your suggestion.

Reviewer 2 Report

Abstract: C3, MIC and MBC should be defined before abbreviating. Use antibactrial instead of antimicrobial 

Introduction: In introdcution, while the authors have given the negative effects of Zn and copper and have advocated the superior effects of silver. But the authors have given less space to silver nanoparticles which is the actual topic of discussion. Moreover, Zn and Cu in inorganic form were found to have negative effects, but the organic forms of these elements are still considered superior as feed additives in animal feed. The authors should give possible review of literature on this aspect rather than giving a huge text citing the ordinary reference

Results: why statistical analysis was not applied on the results?

Results in Tables given have been less discussed while those in figures are more discussed. Table 2 shows just one sample?

Discussion

Is it necessary to revise the objective over here?

While comparing results with other studies now in discussion, why they were ignored in the introduction, and what the gap left were filled by this study? so that the novelty of this study is cleared. 

materials and methods

How the bacterial strains were confirmed?

Author Response

Abstract: C3, MIC and MBC should be defined before abbreviating. Use antibacterial instead of antimicrobial

 We appreciate your comment and according to your suggestion, the abstract has been revised.

Introduction: In introduction, while the authors have given the negative effects of Zn and copper and have advocated the superior effects of silver. But the authors have given less space to silver nanoparticles which is the actual topic of discussion. Moreover, Zn and Cu in inorganic form were found to have negative effects, but the organic forms of these elements are still considered superior as feed additives in animal feed. The authors should give possible review of literature on this aspect rather than giving a huge text citing the ordinary reference.

Thanks for your comment. According to your suggestion, the introduction of the article have been improved.

 Results: why statistical analysis was not applied on the results?

Thanks for your comment. Some statistical analysis has been performed.

Results in Tables given have been less discussed while those in figures are more discussed. Table 2 shows just one sample?

According with your suggestion, we have improved the text regarding the results on the Tables. In addition, we have specified in Table 2 that the tests were carried out in triplicate.

Discussion

 Is it necessary to revise the objective over here?

Yes, you are right; we have removed this aspect from the discussion section.

While comparing results with other studies now in discussion, why they were ignored in the introduction, and what the gap left were filled by this study? so that the novelty of this study is cleared.

We are very grateful to the reviewer for the suggestions made and in order to clarify theses aspects, we have modified the introduction and the discussion adding the novelty of the study. 

Materials and methods:

How the bacterial strains were confirmed?

Thanks for your comment. The text has been clarified according to your suggestion, including the previously performed tests for the confirmation of the strains.

Round 2

Reviewer 2 Report

the authors have revised the paper according to the suggestion and accepted for publication